# Derivatives of Amaryllidaceae Alkaloid Ambelline as Selective Inhibitors of Hepatic Stage of *Plasmodium berghei* Infection In Vitro

**DOI:** 10.3390/pharmaceutics15031007

**Published:** 2023-03-21

**Authors:** Kateřina Hradiská Breiterová, Aneta Ritomská, Diana Fontinha, Jana Křoustková, Daniela Suchánková, Anna Hošťálková, Marcela Šafratová, Eliška Kohelová, Rozálie Peřinová, Rudolf Vrabec, Denise Francisco, Miguel Prudêncio, Lucie Cahlíková

**Affiliations:** 1Secondary Metabolites of Plants as Potential Drugs Research Group, Department of Pharmacognosy and Pharmaceutical Botany, Faculty of Pharmacy, Charles University, Heyrovského 1203, 500 05 Hradec Králové, Czech Republic; 2Prudêncio Lab, Instituto de Medicina Molecular João Lobo Antunes, Faculdade de Medicina, Universidade de Lisboa, Av. Prof. Egas Moniz, Edf. Egas Moniz, 1649-028 Lisboa, Portugal

**Keywords:** alkaloids, Amaryllidaceae, ambelline, cytotoxicity, haemanthamine, hepatic stage, malaria, *Plasmodium*

## Abstract

The incidence rate of malaria and the ensuing mortality prompts the development of novel antimalarial drugs. In this work, the activity of twenty-eight Amaryllidaceae alkaloids (**1–28**) belonging to seven different structural types was assessed, as well as twenty semisynthetic derivatives of the β-crinane alkaloid ambelline (**28a–28t**) and eleven derivatives of the α-crinane alkaloid haemanthamine (**29a–29k**) against the hepatic stage of *Plasmodium* infection. Six of these derivatives (**28h**, **28m**, **28n** and **28r–28t**) were newly synthesized and structurally identified. The most active compounds, 11-*O*-(3,5-dimethoxybenzoyl)ambelline (**28m**) and 11-*O*-(3,4,5-trimethoxybenzoyl)ambelline (**28n**), displayed IC_50_ values in the nanomolar range of 48 and 47 nM, respectively. Strikingly, the derivatives of haemanthamine (**29**) with analogous substituents did not display any significant activity, even though their structures are quite similar. Interestingly, all active derivatives were strictly selective against the hepatic stage of infection, as they did not demonstrate any activity against the blood stage of *Plasmodium* infection. As the hepatic stage is a bottleneck of the plasmodial infection, liver-selective compounds can be considered crucial for further development of the malaria prophylactics.

## 1. Introduction

Malaria is a severe parasitic infection of global relevance caused by unicellular protozoa from the genus *Plasmodium,* transmitted to their mammalian hosts by female *Anopheles* mosquitoes. According to the World Health Organization (WHO), 40% of the world’s population lives in areas at risk of malaria, and in 2020, 241 million cases of this disease were reported, resulting in 627 thousand deaths [1]. There is also a serious economic and social onus on malaria-endemic countries [2,3]. The most important factors related to malaria control are vector control measures, prevention, early diagnosis, and appropriate and effective medication. However, the development of drug resistance in *Plasmodium* spp. contributes a crucial impetus for new potential drug research [4].

There are fourteen subgenera and over two hundred species in the genus *Plasmodium*, but only five species cause malaria in humans—*P. falciparum*, *P. vivax*, *P. ovale*, *P. malariae* and *P. knowlesi* [5,6]. From these, *P. falciparum* infection leads to the most severe cases [5]. The human transmissible *Plasmodium* parasites cycle between two hosts—*Anopheles* mosquitoes and humans. The human part of the lifecycle is divided into two phases—hepatic or liver and the erythrocytic or blood stages of the infection. After the mosquito blood meal, sporozoites, the liver-infective forms of the parasite, are inoculated into the skin of the human host, and they enter the vascular system, eventually reaching the liver. There, each sporozoite traverses a few hepatocytes before productively invading a final one, where it develops into tens of thousands of merozoites, the blood-infective forms of the parasite [7]. Parasites are then released into the bloodstream inside merosomes that burst to release erythrocyte-infective merozoites [8,9,10]. Most merozoites start a cycle of asexual schizogony, but a small part of them starts the sexual cycle—gametocytogenesis [6,11]. Gametocytes may be uptaken by the mosquito vector upon a bloodmeal, where they undergo a series of steps that give rise to new sporozoites, thus completing the parasite’s life cycle. In *P. vivax* and *P. ovale* infections, the liver may harbour hypnozoites, dormant parasites that can cause malaria relapses many weeks to years after the primary infection [6]. The liver stage of infection is clinically silent but is obligatory for all mammalian-infective *Plasmodium* species. Collectively, these traits make this bottleneck of infection the ideal pharmacological target for malaria prophylaxis [12,13]. Furthermore, targeting the liver is the only way to eliminate the dormant forms of *P. vivax* and *P. ovale* parasites and, thus, to achieve a radical cure in the case of these hypnozoite-forming parasite species. Although most of the drugs currently used against malaria are targeted to the erythrocytic stage of infection, primaquine and tafenoquine target liver parasite forms, including hypnozoites [14,15]. Regrettably, primaquine and tafenoquine cannot be used in the treatment of patients who are deficient in glucose-6-phosphate dehydrogenase enzyme (G6PD), as they are at increased risk of developing hemolysis, potentially leading to hemolytic anemia [16,17]. The obligatory nature of the liver stage of *Plasmodium* infection makes it an attractive drug target for many researchers. Although these stages pose several experimental challenges [18], they are promising, as they correspond to a relatively low parasite load in the organism [10,12]. Moreover, new drugs with different mechanisms of action are necessary for the treatment of G6PD-deficient patients.

During the last half-century, *P. falciparum* has developed resistance against several available antimalarial drugs (chloroquine, mefloquine, sulfadoxine-pyrimethamine combination, etc.). Many studies revealed that resistance is developed mainly due to the spontaneous genetic mutation or amplification in key enzymes or transporters [8,19]. Historically, *P. falciparum* resistance occurred in Cambodia, from where it expanded to the Greater Mekong Subregion and the rest of the malaria-affected area. Artemisinin-based combination therapy (ACT) has been used as a first-line treatment for non-severe malaria [4,20,21]. Unfortunately, signs of *P. falciparum* resistance to artemisinin and ACT are beginning to emerge as a serious threat to the whole region and mainly to all WHO malaria control and elimination endeavors [22,23]. In recent years, research attention has shifted to different *Plasmodium* stages and various biochemical pathways of the parasite‘s metabolism, hoping for new potential therapeutics. Possible targets are liver parasites, asexual and sexual blood stages, and even dormant hypnozoites [8]. An alternative to de novo drug development is the repurposing of existing drugs against the new target. This option makes the process of drug discovery and licensing much faster and cheaper due to the previously performed pre-clinical and clinical safety studies [10,24,25].

Forty-nine percent of all drugs and 67% of small-molecule drugs approved between 1981–2019 are natural compounds, their derivatives, or compounds inspired by their structural moieties [26]. This proves that natural compounds and their derivatives are an important source for potential new drugs. This is true also for the treatment of malaria, which initially resorted to the alkaloid quinine (*Cinchona* spp., Rubiaceae), and more recently followed by the discovery of the sesquiterpene lactone artemisinin (*Artemisia annua*, Asteraceae) and its semisynthetic derivatives [8,27]. One of the most interesting groups of bioactive natural compounds are alkaloids, particularly Amaryllidaceae alkaloids (AmA). This group of alkaloids is specific to the Amaryllidaceae family, which consists of over 1600 species in 81 genera [28] and which is one of the most important alkaloidal families, with almost 600 alkaloids isolated and structurally described so far, providing us with a rich source of compounds of various structural types to study [29,30,31]. The biological properties described for these alkaloids include analgesic, anti-inflammatory, antitumor, and cholinesterase-inhibitory activity, in addition to antimalarial activity [32]. Unfortunately, many of these alkaloids are present in low concentrations in plants. Conversely, some AmAs are present in abundant amounts in specific species or cultivars, so they can be used as structural scaffolds for the development of a diversified spectrum of potential drug candidates. Up to date, extracts prepared from 22 various Amaryllidaceae species have been screened for their antiplasmodial activity [30,33,34,35,36]. Except one, all those studies were conducted on avian *P. galinaceum* parasites, as well as on several chloroquine-sensitive (3D7, D-6, D-10, NF54, T9.96), chloroquine-resistant (FAC8, FCR-3, RKL-2, W-2), and multidrug-resistant (Dd2, K-1) strains of *P. falciparum*. Nevertheless, none of them has been tested yet against the liver stage of *Plasmodium* infection [30,33]. Only one study was conducted on *P. berghei* in mice with an interesting activity, which could be caused by a high content of lycorine in the studied fraction of *Crinum jagus* (Thomps) Dandy [35]. From the isolated alkaloids that have already been characterized, acetylcaranine and ambelline showed promising activity against the resistant Dd2 *P. falciparum* strain, with IC_50_ values of 3.5 ± 0.3 μM and 7.3 ± 0.3 μM, respectively [34]. In another study, lycorine displayed an interesting activity against the chloroquine-sensitive 3D7 strain (IC_50_ = 2.5 μM) and the multidrug-resistant K1 strain (IC_50_ = 3.1 μM) [33]. In the present study, the potential antiplasmodial activity was evaluated for twenty-eight structurally diverse AmAs (**1–28**) and thirty-one semisynthetic derivatives of ambelline (**28a–28t**) and haemanthamine (**29a–29k**) with diverse substitutions.

## 2. Materials and Methods

### 2.1. General Experimental Procedures

All solvents were handled according to standard procedures before use. All reagents and catalysts were purchased from a commercial source (Sigma Aldrich, St. Louis, MO, USA) and used without further purification. NMR spectra were recorded in CDCl_3_ on a VNMR S500 (Varian) spectrometer operating at 500 MHz for ^1^H and 125.7 MHz for ^13^C at 25 °C. The coupling constant (*J*) is given in Hz, and the chemical shifts are reported in ppm. ESI-HRMS were obtained with a Waters Synapt G2-Si hybrid mass analyzer of a quadrupole-time-of-flight (Q-TOF) type, coupled to a Waters Acquity I-Class UHPLC system. The EI-MS were obtained on an Agilent 7890A GC 5975 inert MSD operating in EI mode at 70 eV (Agilent Technologies, Santa Clara, CA, USA). A DB-5 column (30 m × 0.25 mm × 0.25 μm, Agilent Technologies, USA) was used. The used temperature program was: 100–180 °C at 15 °C/min, 1 min hold at 180 °C, and 180–300 °C at 5 °C/min and 5 min hold at 300 °C with a detection range of *m/z* 40–600. The injector temperature was 280 °C. The flow-rate of the carrier gas (helium) was 0.8 mL/min, and a split ratio of 1:15 was used. The ESI-MS were obtained using a Waters Autopurification™ HPLC-MS system (Milford, CT, USA). The apparatus consisted of a Waters Sample Manager 2767, a System Fluidics organizer, two Waters 515 HPLC pumps, a Waters 2545 Binary Gradient module, a Waters 2998 Photodiode array detector, and a Waters Acquity qDa detector. The sample was analyzed at ambient temperature using a XSelect**^®^** CSH™ C18 OBD™ reverse phase column (100 mm × 4.6 mm i.d., 5 µm) (Milford, USA). Water with 0.1% formic acid (solvent A) and methanol (MeOH) with 0.1% formic acid (solvent B) were used as mobile phases. The flow rate of the mobile phase was 1 mL/min. The gradient elution program was programmed as follows (*v/v*): 0 min 5% B, 5 min 100% B, 8.5 min 5% B, followed by 1.5 min at initial conditions for re-equilibration. The optimum values of the ESI-MS parameters were: capillary voltage 0.8 kV; probe temperature 600 °C; and cone voltage 15V. LC/MS were recorded across the range of 200–800 *m/z*. For PDA detection, the detector range was set from 190 to 700 nm. LC ESI-MS analyses were carried out in the positive ion mode. All isolated and prepared compounds were either analyzed or purified by TLC on precoated silica gel 60 F254 plates (Merck, Darmstadt, Germany). Compounds on the plates were observed using UV light (λ = 254 and 366 nm) and visualized by spraying with Dragendorff’s reagent. Optical rotation was measured on a P3000 polarimeter in either CHCl_3_ or MeOH.

### 2.2. Amaryllidaceae Alkaloids

All tested AmAs (**1–28**), together with haemanthamine (**29**), were isolated and described in detail during our thorough phytochemical studies performed on *Nerine bowdenii* (W. Watson) [37], *Narcissus poeticus* cv. (Pink Parasol) [38], *Narcissus pseudonarcissus* cv. (Dutch Master) [39], and *Narcissus* cv. (Professor Einstein) [40]. The fresh bulbs of all plant species were obtained from the herbal dealer Lukon Glads (Sadská, Czech Republic). Botanical identification was performed by Prof. L. Opletal. A voucher specimen of each plant is deposited in the Herbarium of the Faculty of Pharmacy in Hradec Králové.

### 2.3. Preparation of Ambelline (***28***) and Haemanthamine (***29***) Derivatives

The elaborated description of the acylation of ambelline (**28**) was previously published by Maříková et al., together with NMR spectra of compounds **28b–28g**, **28i–28l**, and **28o–28q** [41]. The NMR spectra of acetylambelline (**28a**) were previously reported by Viladomat et al. [42]. The description of the acylation of haemanthamine (**29**) was previously published by Kohelová et al. and Peřinová et al. [43,44].

### 2.4. NMR Spectra of Newly Synthesized Derivatives

NMR spectra and HRMS of newly developed semisynthetic derivatives of ambelline can be found in Appendix A. NMR spectra and HRMS of all previously published semisynthetic derivatives can be found in the following references: **28b–28g**, **28i–28l**, and **28o–28q** [41]; **29a–29b**, **29f**, and **29k** [43]; and **29c–29e** and **29g–29j** [44].

#### 2.4.1. 11-*O*-(3,5-Dimethylbenzoyl)ambelline (**28h**)

Yield 67 mg (94%); white powder; [α]^24^_D_ = +20.2 (*c* = 0.139; CHCl_3_); **^1^H NMR** (500 MHz, CDCl_3_) *δ*: 7.22 (2H, s), 7.12 (1H, s), 6.59 (1H, d, *J* = 10.0 Hz), 6.57 (1H, s), 6.08 (1H, dd, *J* = 10.0 Hz, *J* = 5.2 Hz), 5.86 (1H, d, *J* = 1.5 Hz), 5.80 (1H, d, *J* = 1.5 Hz), 5.33 (1H, dd, *J* = 8.0 Hz, *J* = 3.8 Hz), 4.37 (1H, d, *J* = 17.5 Hz), 4.00 (3H, s), 3.93 (1H, overlap, d, *J* = 17.5 Hz), 3.93–3.87 (2H, overlap, m), 3.53 (1H, ddd, *J* = 13.7 Hz, *J* = 4.0 Hz, *J* = 1.7 Hz), 3.37 (3H, s), 2.80 (1H, ddd, *J* = 14.2 Hz, *J* = 4.0 Hz, *J* = 1.7 Hz), 2.28 (6H, s), 2.21–2.16 (1H, m), 1.79 (1H, td, *J* = 13.7 Hz, *J* = 4.0 Hz); **^13^C NMR** (125.7 MHz, CDCl_3_) *δ*: 166.4, 147.8, 140.6, 137.7, 134.5, 133.9, 133.8, 131.2, 129.7, 127.2, 126.6, 117.3, 100.5, 99.6, 87.3, 72.2, 63.0, 59.8, 59.1, 58.6, 56.5, 47.5, 28.7, 21.0; **ESI-HRMS**: *m/z* calcd for C_27_H_30_NO_6_^+^ [M+H]^+^ 464.2068, found 464.2067.

#### 2.4.2. 11-*O*-(3,5-Dimethoxybenzoyl)ambelline (**28m**)

Yield 58 mg (75%); yellow powder; [α]^24^_D_= +32.4 (*c* = 0.15; CHCl_3_); **^1^H NMR** (500 MHz, CDCl_3_) δ: 6.80 (2H, d, *J* = 2.5 Hz), 6.60–6.55 (3H, m), 6.07 (1H, dd, *J* = 10.3 Hz, *J* = 4.9 Hz), 5.81 (1H, d, *J* = 1.4 Hz), 5.77 (1H, d, *J* = 1.4 Hz), 5.31 (1H, dd, *J* = 7.8 Hz, *J* = 3.9 Hz), 4.35 (1H, d, *J* = 17.1 Hz), 3.99 (3H, s), 3.94–3.85 (3H, m), 3.73 (6H, s), 3.50 (1H, dd, *J* = 13.6 Hz, *J* = 3.9 Hz), 3.35 (3H, s), 2.82–2.75 (1H, m), 2.17 (1H, dd, *J* = 14.2 Hz, *J* = 4.4 Hz), 1.78 (1H, td, *J* = 13.6 Hz, *J* = 3.9 Hz); **^13^C NMR** (125.7 MHz, CDCl_3_) δ: 165.9, 160.4, 147.7, 140.6, 133.9, 133.7, 131.6, 131.1, 126.7, 117.3, 106.6, 106.3, 100.4, 99.4, 87.8, 72.1, 63.0, 59.9, 59.1, 58.7, 56.5, 55.3, 47.2, 28.6; **ESI-HRMS**: *m/z* calcd for C_27_H_30_NO_8_^+^ [M+H]^+^ 496.1966, found 496.1977.

#### 2.4.3. 11-*O*-(3,4,5-Trimethoxybenzoyl)ambelline (**28n**)

Yield 17 mg (36%); white powder; [α]^24^_D_= +83.3 (*c* = 0.12; CHCl_3_); **^1^H NMR** (500 MHz, CDCl_3_) δ: 6.94 (2H, s), 6.62 (1H, s), 6.58 (1H, d, *J* = 10.0 Hz), 6.08 (1H, dd, *J* = 10.0 Hz, *J* = 5.1 Hz), 5.80 (2H, s), 5.31 (1H, dd, *J* = 7.9 Hz, *J* = 3.9 Hz), 4.36 (1H, d, *J* = 17.6 Hz), 3.98 (3H, s), 3.97–3.87 (3H, m), 3.86 (3H, s), 3.81 (6H, s), 3.49 (1H, dd, *J* = 13.8 Hz, *J* = 3.9 Hz), 3.37 (3H, s), 2.77 (1H, ddd, *J* = 14.2 Hz, *J* = 3.9 Hz, *J* = 1.4 Hz), 2.22–2.16 (1H, m), 1.80 (1H, td, *J* = 13.8 Hz, *J* = 3.9 Hz); **^13^C NMR** (125.7 MHz, CDCl_3_) δ: 165.8, 152.7, 147.6, 142.1, 140.7, 134.1, 133.7, 131.0, 126.9, 124.7, 117.4, 106.5, 100.5, 99.4, 87.9, 72.2, 63.0, 60.8, 60.1, 59.1, 58.8, 56.6, 55.9, 47.0, 28.7; **ESI-HRMS**: *m/z* calcd for C_28_H_32_NO_9_^+^ [M+H]^+^ 526.2072, found 526.2077.

#### 2.4.4. 11-*O*-(4-Methyl-3-nitrobenzoyl)ambelline (**28r**)

Yield 76 mg (100%); pale yellow oil; [α]^24^_D_= +78.7 (*c* = 0.12; CHCl_3_); **^1^H NMR** (500 MHz, CDCl_3_) δ: 7.97–7.95 (1H, m), 7.88 (1H, dd, *J* = 7.8 Hz, *J* = 1.5 Hz), 7.36 (1H, d, *J* = 7.8 Hz), 6.56 (1H, d, *J* = 10.0 Hz), 6.50 (1H, s), 6.09 (1H, dd, *J* = 10.0 Hz, *J* = 5.2 Hz), 5.91–5.89 (1H, m), 5.87–5.86 (1H, m), 5.40 (1H, dd, *J* = 8.0 Hz, *J* = 3.7 Hz), 4.38 (1H, d, *J* = 17.5 Hz), 4.02 (3H, s), 3.94 (1H, overlap, d, *J* = 17.5 Hz), 3.94–3.88 (2H, overlap, m), 3.55 (1H, dd, *J* = 13.8 Hz, *J* = 3.9 Hz), 3.37 (3H, s), 2.87–2.81 (1H, m), 2.62 (3H, s), 2.23–2.16 (1H, m), 1.78 (1H, td, *J* = 13.8 Hz, *J* = 3.9 Hz); **^13^C NMR** (125.7 MHz, CDCl_3_) δ: 164.2, 149.0, 148.1, 140.7, 138.5, 134.1, 133.29, 133.25, 132.9, 130.9, 129.2, 126.9, 125.7, 117.0, 100.8, 99.0, 88.1, 72.1, 63.2, 59.8, 59.2, 58.6, 56.6, 47.6, 28.8, 20.6; **ESI-HRMS**: *m/z* calcd for C_26_H_27_N_2_O_8_^+^ [M+H]^+^ 495.1762, found 495.1771.

#### 2.4.5. 11-*O*-(2-Chloro-4-nitrobenzoyl)ambelline (**28s**)

Yield 91 mg (100%); pale yellow oil; [α]^24^_D_= −8.3 (*c* = 0.10; CHCl_3_); **^1^H NMR** (500 MHz, CDCl_3_) δ: 8.21 (1H, d, *J* = 2.1 Hz), 7.99 (1H, dd, *J* = 8.3 Hz, *J* = 2.1 Hz), 7.29 (1H, d, *J* = 8.3 Hz), 6.65 (1H, d, *J* = 10.0 Hz), 6.47 (1H, s), 6.10 (1H, dd, *J* = 9.8 Hz, *J* = 5.4 Hz), 5.82–5.79 (2H, m), 5.34 (1H, dd, *J* = 7.8 Hz, *J* = 3.5 Hz), 4.36 (1H, d, *J* = 17.6 Hz), 3.98 (3H, s), 3.94–3.85 (3H, m), 3.51 (1H, dd, *J* = 13.6 Hz, *J* = 3.5 Hz), 3.36 (3H, s), 2.94–2.87 (1H, m), 2.20–2.12 (1H, m), 1.75 (1H, td, *J* = 13.6 Hz, *J* = 3.5 Hz); **^13^C NMR** (125.7 MHz, CDCl_3_) δ: 163.6, 149.3, 147.9, 140.7, 135.0, 134.9, 134.0, 133.3, 132.1, 131.3, 126.8, 125.9, 121.0, 117.6, 100.6, 99.2, 89.5, 72.1, 63.6, 59.6, 59.1, 58.5, 56.5, 47.9, 28.8; **ESI-HRMS**: *m/z* calcd for C_25_H_24_ClN_2_O_8_^+^ [M+H]^+^ 515.1216, found 515.1227.

#### 2.4.6. 11-*O*-(4-Chloro-3-nitrobenzoyl)ambelline (**28t**)

Yield 52 mg (59%); yellow powder; [α]^24^_D_= +27.2 (*c* = 0.24; CHCl_3_); **^1^H NMR** (500 MHz, CDCl_3_) δ: 7.88 (1H, dd, *J* = 8.3 Hz, *J* = 2.0 Hz), 7.84 (1H, d, *J* = 2.0 Hz), 7.57 (1H, d, *J* = 8.3 Hz), 6.54 (1H, d, *J* = 10.1 Hz), 6.47 (1H, s), 6.14–6.06 (1H, m), 5.90–5.86 (2H, m), 5.32 (1H, dd, *J* = 8.1 Hz, *J* = 3.6 Hz), 4.37 (1H, d, *J* = 17.6 Hz), 4.02 (3H, s), 3.97–3.85 (3H, m), 3.56 (1H, dd, *J* = 13.7 Hz, *J* = 4.2 Hz), 3.37 (3H, s), 2.87–2.81 (1H, m), 2.23–2.16 (1H, m), 1.76 (1H, td, *J* = 13.7 Hz, *J* = 4.2 Hz); **^13^C NMR** (125.7 MHz, CDCl_3_) δ: 163.3, 148.1, 147.6, 140.8, 134.1, 133.4, 133.2, 132.0, 131.7, 130.7, 129.8, 127.1, 126.5, 116.9, 100.9, 98.8, 88.4, 72.1, 63.2, 59.7, 59.2, 58.6, 56.6, 47.6, 28.7; **ESI-HRMS**: *m/z* calcd for C_25_H_24_ClN_2_O_8_^+^ [M+H]^+^ 515.1216, found 515.1224.

### 2.5. In Vitro Activity against P. berghei—Hepatic Stages

Activity against the hepatic stage of *P. berghei* infection was assessed using the human hepatoma cell line Huh-7 (deposited at the Japanese Collection of Research Bioresources under JCRB0403). The protocol was previously comprehensively described by Prudêncio et al. [45] and by Ploemen et al. [46]. Briefly, cell culture medium consisted of 1640 RPMI medium supplemented with 10% *v/v* fetal calf serum, 1% *v/v* L-glutamine, 1% *v/v* nonessential amino acids, 1% *v/v* penicillin/streptomycin, and 10 mM HEPES, pH 7. Cells were seeded in 96-well plates at a density of 1 × 10^4^ cells per well in 100 µL of culture medium and incubated at 37 °C with 5% CO_2_ for approximately 18 h. Tested compounds were dissolved in DMSO to 10 mM stock solutions and then diluted with infection medium (culture medium enriched with 50 µg/mL of gentamicin and 0.8 µg/mL of amphotericin B) to concentrations of 10 and 1 µM for screening and various concentrations ranging from 30 to 0.0001 µM for IC_50_ determination, respectively. The culture medium was substituted by infection medium with diluted compounds 1 hour in advance of infection. Luciferase-expressing *P. berghei* sporozoites (PbA-LuciGFPcon spz) were freshly obtained through the dissection of salivary glands from infected female *Anopheles stephensi* mosquitoes. Sporozoites were added to the cells at 1:1 ratio, i.e., 1 × 10^4^ sporozoites were added per well, and then subjected to centrifugation at 1800× *g* for 5 min. The plate was then incubated for 46 h at 37 °C with 5% CO_2_, at which time Huh-7 cell confluency was indirectly assessed by the alamarBlue assay (Invitrogen, Waltham, MA, USA), as reported in the manufacturer’s protocol. Parasite load was assessed 48 h post-infection by a bioluminescence assay, as reported by the manufacturer (Biotium, Fremont, CA, USA). GraphPad Prism 8 (GraphPad Software, Inc., La Jolla, CA, USA) was used to obtain IC_50_ values using nonlinear regression analysis to fit the dose-response curves normalized results.

### 2.6. In Vitro Activity against P. falciparum Blood Stages

Activity against the blood stage of *P. falciparum* infection was assessed by flow cytometry analysis following the incubation of either test compounds or the drug vehicle (DMSO) with ring-stage synchronized parasite cultures, as previously described [47]. Stock solutions of the test compounds, as well as of the positive control compound chloroquine, were prepared in DMSO and diluted to test concentrations in complete malaria culture medium (CMCM), i.e., 1640 RPMI medium supplemented with 25 mM HEPES, 2.4 mM L-glutamine, 50 µg/mL gentamicin, 0.5% *w*/*v* Albumax, 11 mM glucose, 1.47 mM hypoxanthine, and 37.3 mM NaHCO_3_. Samples at tested concentration were then incubated with ring-stage synchronized cultures of *P. falciparum* NF54 at 2.5% hematocrit and at approximately 1% parasitemia in 96-well plates at 37 °C in a 5% CO_2_ and 5% O_2_ atmosphere. Forty-eight hours post incubation, parasite load was assessed by flow cytometry, upon staining with the DNA-specific dye SYBR green I. Approximately 100,000 events were analyzed in each flow cytometry measurement.

### 2.7. Cytotoxicity Assessment In Vitro (MTT Assay)

HepG2 cells of human hepatocellular carcinoma (ATCC HB-8065; passage 20–25), purchased from Health Protection Agency Culture Collections (ECACC, Salisbury, UK), were cultured in Minimum Essential Medium Eagle supplemented with 10% *v/v* fetal bovine serum and 1% *v/v* L-glutamine solution (Sigma-Aldrich, St. Louis, MO, USA) at 37 °C in a humidified atmosphere containing 5% CO_2_. For passaging, the cells were treated with trypsin/EDTA (Sigma-Aldrich, St. Louis, MO, USA) at 37 °C and then harvested. For the cytotoxicity evaluation, the cells treated with the test substances were used, while untreated HepG2 cells served as a control group. The cells were seeded in a 96-well plate at a density of 1 × 10^4^ cells per well and incubated for 24 h. All tested compounds were dissolved in DMSO to prepare 10 mM stock solutions and diluted to the desired concentration just before the cell treatment. Cytotoxicity of all studied derivatives was screened at 10 μM concentration, and all samples were measured in triplicates. Positive and negative controls were also included. Plates were incubated for 24 h in a humidified atmosphere containing 5% CO_2_ at 37 °C. After the incubation, a solution of thiazolyl blue tetrazolium bromide (Sigma-Aldrich, St. Louis, MO, USA) in RPMI 1640 medium without phenol red (BioTech, Prague, Czech Republic) was added and incubated for 3 h in a humidified atmosphere containing 5% CO_2_ at 37 °C. Afterwards, formazan crystals were dissolved in DMSO, and the absorbance of samples was recorded at 570 nm (Synergy Neo2 Multi-Mode Reader NEO2SMALPHAB; BioTek, Winooski, VT, USA).

## 3. Results and Discussion

### 3.1. Studied Alkaloids and Their Semisynthetic Derivatives

The main aim of this study was to assess the antiplasmodial activity of AmA and their semisynthetic derivatives. Eight AmAs (specifically narwedine, homolycorine, masonine, lycoramine, galanthamine, oduline, haemanthamine, and hippeastrine) were tested in the initial experiment, and the results were published by Šafratová et al. [38]. Herein, twenty-eight other AmA (Figure 1) and thirty-one semisynthetic derivatives of ambelline (**28**, Figure 2) and haemanthamine (**29**, Figure 3) were tested. The selected AmAs can be divided into seven groups regarding the structural types that they belong to. Galanthamine-type alkaloids are represented by chlidanthine (**1**), lycoraminone (**2**), and norlycoramine (**3**); lycorine-type by caranine (**4**), acetylcaranine (**5**), lycorine (**6**), 1-*O*-acetyllycorine (**7**), galanthine (**8**), 9-*O*-demethylgalanthine (**9**), and norpluviine (**10**); lycorenine (**11**), O-ethyllycorenine (**12**), eugenine (**13**), 9-*O*-demethylhomolycorine (**14**), and tetrahydromasonine (**15**) belong to the homolycorine-type alkaloids; the tazettine type is represented by a single alkaloid tazettine (**16**), as well as the montanine type by pancracine (**17**); haemanthidine (**18**), hamayne (**19**), epimaritidine (**20**), 9-*O*-demethylmaritidine (**21**), and seco-isopowellaminone (**22**) are haemanthamine (α-crinane) structural type alkaloids, and buphanamine (**23**), 1-*O*-acetylbulbisine (**24**), undulatine (**25**), crinine (**26**), buphanisine (**27**), and ambelline (**28**) are representatives of the crinine (β-crinane) structural type of AmA. The last mentioned alkaloid—ambelline—served as a template structure for the preparation of semisynthetic derivatives (**28a–28t**). Moreover, eleven previously published esters of haemanthamine (**29a–29k**) [43,44] with substituents analogous to the selected esters of ambelline were also studied. The antiplasmodial effect of haemanthamine (**29**) itself was tested previously, but without notable activity [38]. This paper discusses the relationship between structure and activity (SAR) against the hepatic stage of *Plasmodium* infection for the crinine (β-crinane) and haemanthamine (α-crinane) structure type of AmA, together with a discussion of the importance of benzoyl moiety substitution.

### 3.2. Synthesis of Ambelline (***28***) and Haemanthamine (***29***) Derivatives

A sufficient amount of the alkaloid ambelline (**28**) was isolated from the fresh bulbs of *Nerine bowdenii* (W. Watson), while haemanthamine (**29**) was obtained from the fresh bulbs of *Narcissus pseudonarcissus* cv. (Dutch Master), as stated in our preceding studies [37,39]. The purity of both template alkaloids (≥95%) was determined by HRMS and NMR analyses. For the preparation of ambelline esters (**28a–28t**), we proceeded according to the schemes in Figure 2 and as previously described [41]. The synthesis of haemanthamine esters (**29a–29k**) was thoroughly described by Kohelová et al. [43] and Peřinová et al. [44] in our preceding studies. Structures of the tested derivatives can be found in Figure 3. Haemanthamine derivatives chosen for this study were 11-*O*-propionylhaemanthamine (**29a**, substituents are analogous to those of compound **28b**); 11-*O*-isobutanoylhaemanthamine (**29b**, analogous to **28c**); 11-*O*-(3-methylbenzoyl)haemanthamine (**29c**, corresponding to **28f**); 11-*O*-(4-methylbenzoyl)haemanthamine (**29d**, analogous to **28g**); 11-*O*-(2-methoxybenzoyl)haemanthamine (**29e**, analogous to **28i**); 11-*O*-(3-methoxybenzoyl)haemanthamine (**29f**, substitution corresponds to **28j**); 11-*O*-(4-methoxybenzoyl)haemanthamine (**29g**, substituents same as in **28k**); 11-*O*-(3,4-dimethoxybenzoyl)haemanthamine (**29h**, refers to **28l**); 11-*O*-(3,5-dimethoxybenzoyl)haemanthamine (**29i**, analogous to **28m**); 11-*O*-(3-nitrobenzoyl)haemanthamine (**29j**, corresponds to **28p**) and 11-*O*-(4-nitrobenzoyl)haemanthamine (**29k**, related to **28q**). Structures and purity of all newly synthetized derivatives were determined and confirmed by either LC-MS or GC-MS, by HRMS, one-dimensional and two-dimensional NMR, and optical rotation analyses. Data obtained from these analyses were completely consistent with the suggested structures. Similar to our previous reports focused on ambelline/haemanthamine derivatives and their ability to inhibit human cholinesterases associated with Alzheimer’s disease treatment [41,43,44], the substituted aromatic acyl moiety was shown to be essential for the increased antiplasmodial activity in comparison to either ambelline itself or its aliphatic esters.

### 3.3. Antiplasmodial Activity of Selected Alkaloids and Derivatives

Because the antiplasmodial/antimalarial activity of several AmAs has been reported in the literature [36], twenty-eight AmA belonging to various structural types were screened for activity against the hepatic stage of infection caused by luciferase-expressing *P. Berghei* parasites. None of the selected alkaloids exhibited any relevant activity against *P. Berghei* hepatic stages. Some of the screened ambelline derivatives demonstrated promising activity, but, on the other side, the activity of haemanthamine derivatives was lower (Figure 4 and Figure 5). Ambelline derivatives that displayed activity up to 5 μM were selected for further IC_50_ determination. Compound **28t** was also included in this list for comparison purposes, even though it showed lower activity than that of the other compounds. As depicted in Table 1, aliphatic esters of ambelline (**28a–c**) did not display any significant activity—except 11-*O*-pentanoylambelline (**28d**), which showed mild activity with an IC_50_ = 4.08 ± 1.78 μM. Conversely, most aromatic esters of ambelline (**28e–r**) showed more compelling results with IC_50_ values, ranging from 0.047 (**28n**) to 2.40 μM (**28i**). Although compounds **28s** and **28t** exhibited some antiplasmodial activity in the initial screening, this activity was not dose-dependent, which did not allow for the determination of their IC_50_. Interestingly, these are the only derivatives with chlorine substitution in their structure, as they are 11-*O*-(2-chloro-4-nitrobenzoyl)ambelline (**28s**) and 11-*O*-(4-chloro-3-nitrobenzoyl)ambelline (**28t**), respectively. All the aromatic esters showed higher activity against *P. berghei* liver stages than the used standard, primaquine (IC_50_ = 5.74 ± 0.86 μM). The most promising activity was displayed by 11-*O*-(3,4,5-trimethoxybenzoyl)ambelline (**28n**) and 11-*O*-(3,5-dimethoxybenzoyl) ambelline (**28m**), with IC_50_ values in the nanomolar range—0.047 ± 0.000 µM and 0.048 ± 0.014 µM, respectively. Both of these compounds showed certain cytotoxic potential in screening concentrations, but, as shown in Figure 6, the cell viability was not affected at antiplasmodial-active concentrations. Other remarkably active derivatives were 11-*O*-(3,5-dimethylbenzoyl)ambelline (**28h**, IC_50_ = 0.100 ± 0.018 µM), 11-*O*-(3,4-dimethoxybenzoyl) ambelline (**28l**, IC_50_ = 0.147 ± 0.018 µM), and 11-*O*-(3-methoxybenzoyl)ambelline (**28j**, IC_50_ = 0.261 ± 0.090 µM). Compounds that presented the highest activity towards the hepatic stage of *P. berghei* infection were further tested for their potential impact on the subsequent blood stage of infection. None of the studied compounds displayed activity against the human-infective parasite *P. falciparum* blood stages at the tested concentrations compared to the used standard, chloroquine (CQ; Figure 7). These results suggest a liver stage-specific activity for these compounds, which may result from their action on a parasite target or host factor that is limited to this stage of infection, and this supports their further development as potential prophylactic agents.

### 3.4. Structure-Activity Relationships

Ambelline (28) has a 1,2,3,4,5-pentasubstituted benzene ring and belongs to the β-crinane type of AmA, while haemanthamine (29) has a 1,2,4,5-tetrasubstituted benzene ring and belongs to the α-crinane type of AmA. Both alkaloids have a *S*-configuration at C-11. Since neither haemanthamine nor ambelline showed inhibition, drawing a conclusion about their SAR might be misleading. However, some prepared derivatives of ambelline were shown to be potent inhibitors of the hepatic stage of *Plasmodium* infection (Figure 4), whereas the same esters of haemanthamine (Figure 5) did not display such activity. Thus, the spatial orientation of ambellines with 11-*O*-substitution on the β-crinane ethylene bridge is linked with the antiplasmodial activity. The most favorable substitution of ambelline’s benzoyl ester seems to be related to the C-3 together with the C-5 position of the benzoyl moiety. The 3,5-disubstitution of benzoyl with electron-donating groups (EDG: methyl **28h** or methoxy **28m**) significantly improved activity. Surprisingly, this EDG influence does not apply to the 3,5-diethoxybenzoyl derivative (**28o**), probably because of the steric effect of a longer alkyl chain of ether. Simply speaking, EDG-monosubstituted benzoyl derivatives were weaker inhibitors, as well as derivatives with an electron-withdrawing substitution (EWG: nitro group and chlorine **28p–t**). The only tested derivative (**28n**) with a higher benzoyl substitution by three methoxy groups showed significant inhibition, comparable to that of the 3,5-disubstituted derivative **28m**.

### 3.5. Cytotoxicity of Tested Compounds

Cytotoxicity of all isolated AmA and derivatives was screened on a set of ten various cell lines in our previous studies [32,38,40,48]. However, this set did not contain any liver-derived cell line. Since a crucial bottleneck of the *Plasmodium* life cycle takes place in the liver of the host, and, as stated above, ambelline esters showed exclusive inhibition of the plasmodial hepatic stage, and the evaluation of the hepatotoxic potential of the studied compounds was needed. The cytotoxicity of all studied compounds was indicatively screened during the antiplasmodial activity evaluation using the alamarBlue assay to avoid the false positivity associated with the cytotoxic effect of the tested compounds. In addition, a more thorough evaluation of the cytotoxic potential of the investigated compounds was carried out employing the MTT method. All compounds were screened at 10 μM using the hepatocellular carcinoma HepG2 cell line as a commonly used model for hepatotoxicity evaluation of xenobiotics and drug candidates. None of the screened compounds in the tested concentration (10 μM) lowered the cell viability below 50%, except for compound **28t**, which decreased the viability by up to 15% compared to the non-treated control (Figure 8, Table 1). As mentioned above, this compound was excluded from the IC_50_ determination because of its dose-independent activity.

## 4. Conclusions

Twenty-eight Amaryllidaceae alkaloids belonging to seven structural types were shown to lack activity against the hepatic stage of *P. berghei* infection. However, since two alkaloids, ambelline (**28**) and haemanthamine (**29**), were isolated in abundant amounts, the preparation of semisynthetic derivatives led to a SAR study with respect to their antiplasmodial activity. Consequently, several aromatic esters of ambelline have displayed specific activity against the hepatic stage of *P. berghei* infection. Regarding the structure–activity relationship, the spatial orientation of the ethylene bridge (β-crinane type AmA), together with the 11-*O*-substitution, are responsible for the antiplasmodial activity, since haemanthamine derivatives did not show comparable inhibition. The most significant activity, shown by 11-*O*-(3,5-dimethoxybenzoyl)ambelline (**28m**) and 11-*O*-(3,4,5-trimethoxybenzoyl)ambelline (**28n**), is apparently associated with the 3,5-dimethoxy substitution pattern of the ambelline’s benzoyl ester. Additionally, except for **28t**, none of the evaluated compounds was significantly cytotoxic to the HepG2 cell line at 10 μM. This study demonstrated that ambelline derivatives are worthy candidates for further antiplasmodial studies, especially in terms of expanding the portfolio of drugs that can be used to eradicate dormant forms of parasites.

## Figures and Tables

**Figure 1 pharmaceutics-15-01007-f001:**
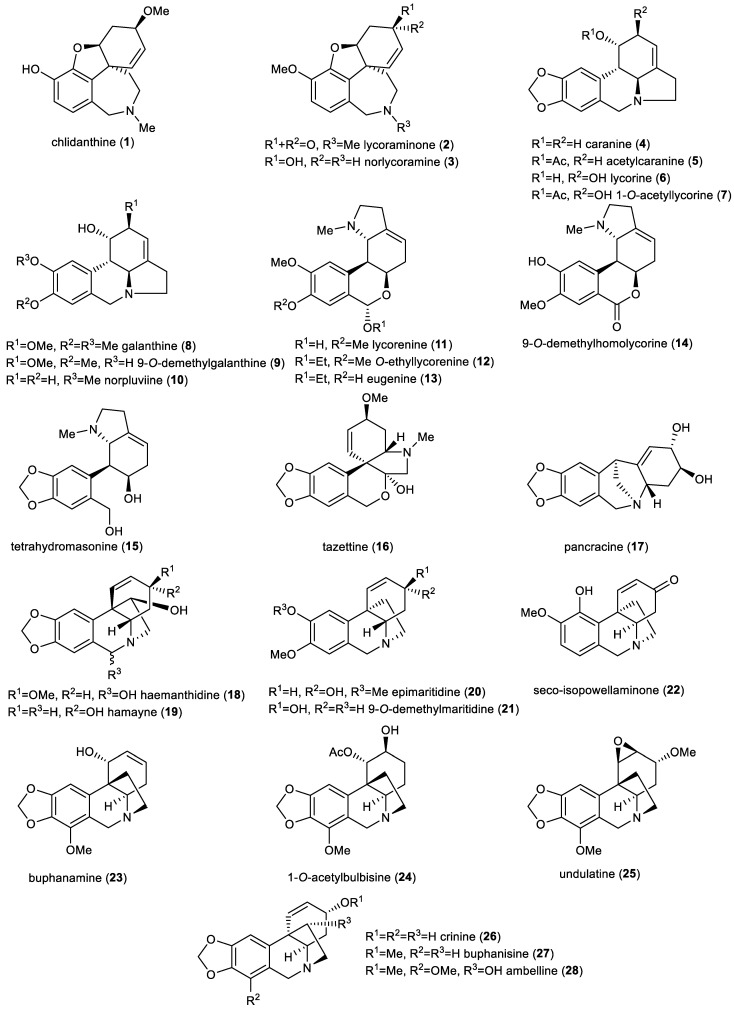
Amaryllidaceae alkaloids tested against the hepatic stage of Plasmodium berghei *infection* in vitro.

**Figure 2 pharmaceutics-15-01007-f002:**
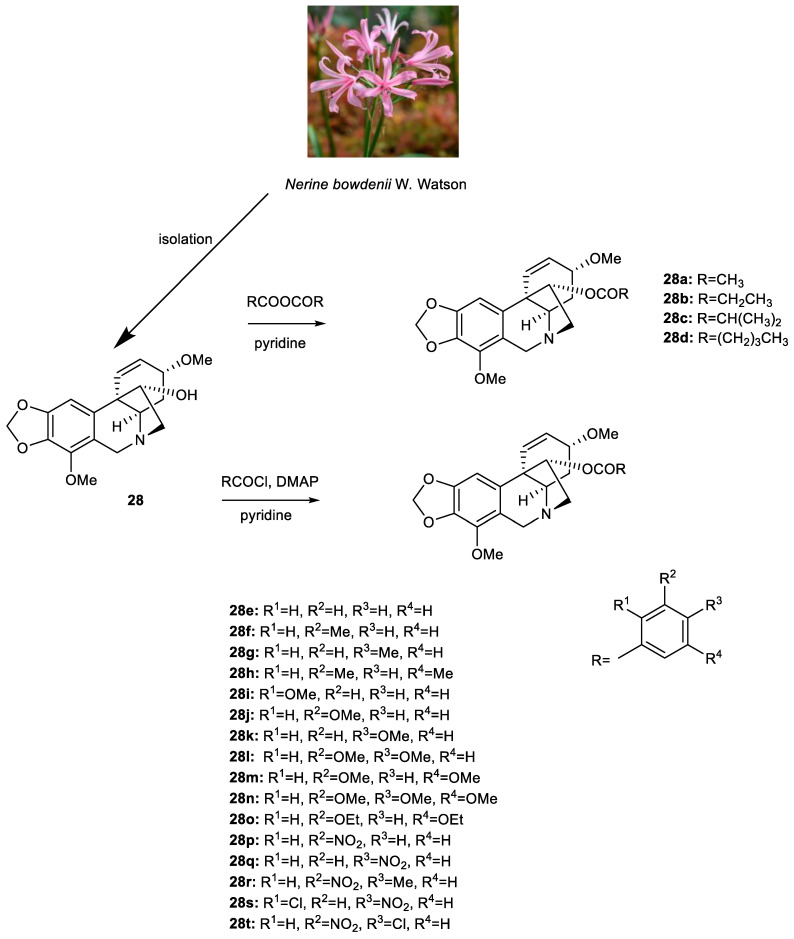
Synthesis of aliphatic (**28a–d**) and aromatic (**28e–t**) esters of ambelline.

**Figure 3 pharmaceutics-15-01007-f003:**
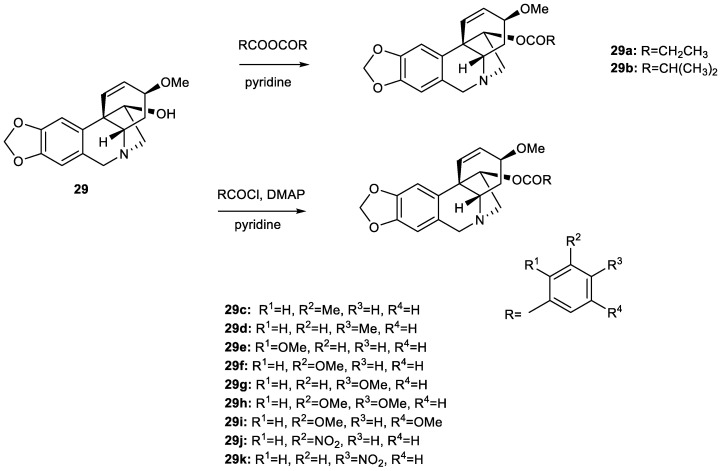
Overview of tested aliphatic and aromatic esters of haemanthamine (**29**).

**Figure 4 pharmaceutics-15-01007-f004:**
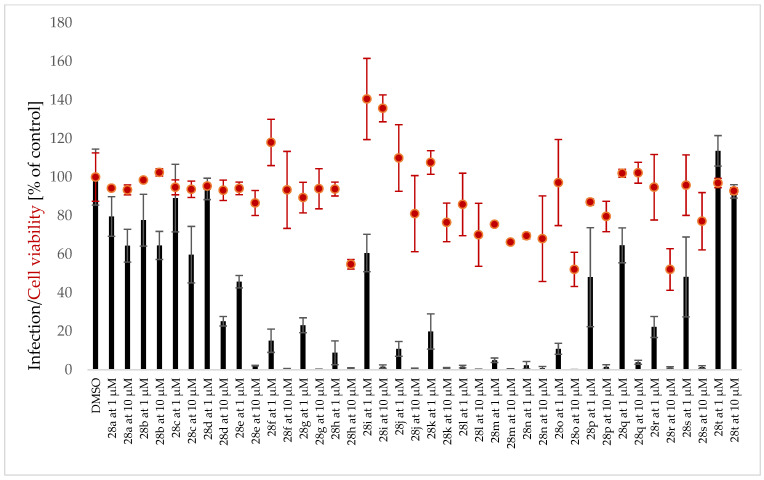
Screening of inhibitory activity of ambelline derivatives against the hepatic stage of Plasmodium berghei in Huh-7 cell line.

**Figure 5 pharmaceutics-15-01007-f005:**
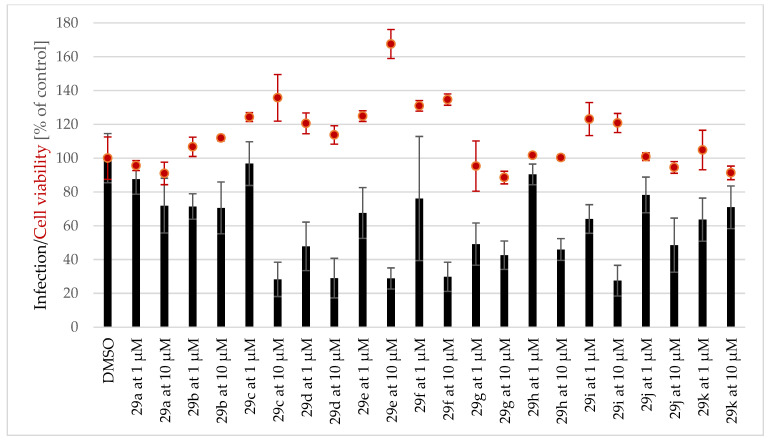
Screening of inhibitory activity of haemanthamine derivatives against the hepatic stage of Plasmodium berghei in Huh-7 cell line.

**Figure 6 pharmaceutics-15-01007-f006:**
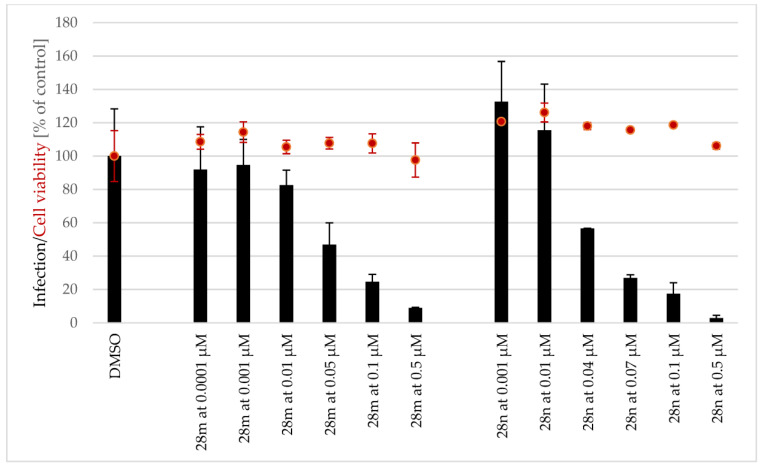
Inhibition of hepatic stage of *Plasmodium* berghei in Huh-7 cell line—IC_50_ determination of two the most active compounds—**28m** and **28n**.

**Figure 7 pharmaceutics-15-01007-f007:**
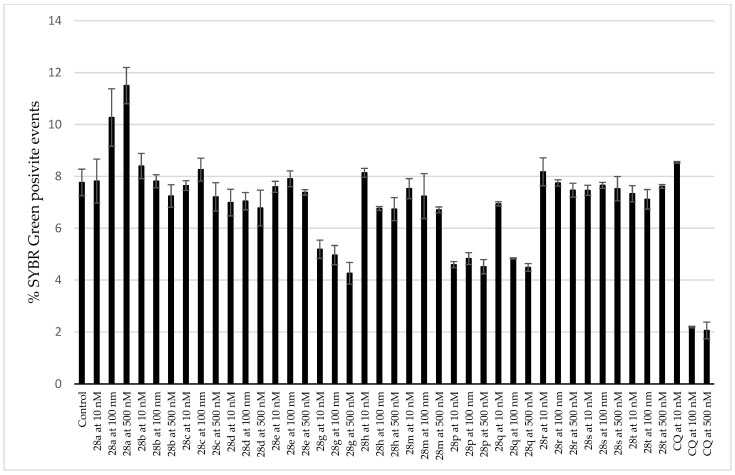
Screening of blood stage inhibitory activity (*Plasmodium* falciparum ring-stage in blood).

**Figure 8 pharmaceutics-15-01007-f008:**
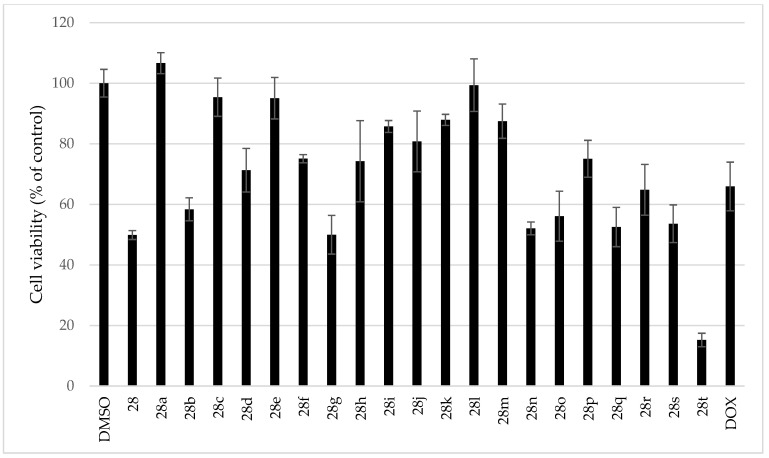
Screening of cytotoxic potential of ambelline derivatives at 10 μM concentration on HepG2 cell line.

**Table 1 pharmaceutics-15-01007-t001:** Activity of ambelline derivatives against the hepatic stage of malaria caused by *P. berghei* and their cytotoxic potential.

Compound	IC_50_ *P. berghei* [µM]	HepG2 [%] ^1^ (c = 10 µM)
Ambelline (**28**)	>10	50 ± 1
11-*O*-Acetylambelline (**28a**)	>10	107 ± 3
11-*O*-Propionylambelline (**28b**)	>10	58 ± 4
11-*O*-Isobutanoylambelline (**28c**)	>10	95 ± 6
11-*O*-Pentanoylambelline (**28d**)	4.08 ± 1.78	71 ± 7
11-*O*-Benzoylambelline (**28e**)	0.726 ± 0.126	95 ± 7
11-*O*-(3-Methylbenzoyl)ambelline (**28f**)	0.317 ± 0.118	75 ± 1
11-*O*-(4-Methylbenzoyl)ambelline (**28g**)	0.402 ± 0.081	50 ± 6
11-*O*-(3,5-Dimethylbenzoyl)ambelline (**28h**)	0.100 ± 0.018	74 ± 13
11-*O*-(2-Methoxybenzoyl)ambelline (**28i**)	2.40 ± 0.77	86 ± 2
11-*O*-(3-Methoxybenzoyl)ambelline (**28j**)	0.261 ± 0.090	81 ± 10
11-*O*-(4-Methoxybenzoyl)ambelline (**28k**)	0.451 ± 0.007	88 ± 2
11-*O*-(3,4-Dimethoxybenzoyl)ambelline (**28l**)	0.147 ± 0.018	99 ± 9
11-*O*-(3,5-Dimethoxybenzoyl)ambelline (**28m**)	0.048 ± 0.014	87 ± 6
11-*O*-(3,4,5-Trimethoxybenzoyl)ambelline (**28n**)	0.047 ± 0.000	52 ± 2
11-*O*-(3,5-Diethoxybenzoyl)ambelline (**28o**)	0.316 ± 0.013	56 ± 8
11-*O*-(3-Nitrobenzoyl)ambelline (**28p**)	1.53 ± 0.22	75 ± 6
11-*O*-(4-Nitrobenzoyl)ambelline (**28q**)	1.20 ± 0.18	53 ± 6
11-*O*-(4-Methyl-3-nitrobenzoyl)ambelline (**28r**)	0.410 ± 0.054	65 ± 8
11-*O*-(2-Chloro-4-nitrobenzoyl)ambelline (**28s**)	not tested	54 ± 6
11-*O*-(4-Chloro-3-nitrobenzoyl)ambelline (**28t**)	not tested	15 ± 2
Primaquine ^2^	5.74 ± 0.86	-
Doxorubicine ^2^	-	66 ± 8

^1^ % of cell viability compared to non-treated control (100%); ^2^ *Reference drug.*

## Data Availability

The data presented in this study are available within the article or Appendix A.

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
