# Peer review of "Derivatives of Amaryllidaceae Alkaloid Ambelline as Selective Inhibitors of Hepatic Stage of Plasmodium berghei Infection In Vitro"

_pharmaceutics, 2023, doi:10.3390/pharmaceutics15031007_

Round 1

Reviewer 1 Report

The paper on anti-Plasmodium activity of selected Amaryllidaceae alkaloids is very interesting and well-written. However, it needs a major revision as suggested below.

In 2.2. Amaryllidaceae Alkaloids, please indicate origin of country for the mentioned plant species as well as plant parts used for isolation. In the same section, “thorough” should be corrected as “through”

2.4. NMR Spectra of Newly Developed Derivatives should be corrected as 2.4. NMR Spectra of Newly Synthesized Derivatives

Line 264, “Test concentrations were then incubated” should be modified as concentrations cannot be incubated. It might be suggested as “samples at tested concentrations”.

“3.1. Studied Alkaloids and Semisynthetic Derivatives” should be corrected as “3.1. Studied Alkaloids and Their Semisynthetic Derivatives”

Line 302, what is single tazettine type alkaloid? Indicate here please.

In Figure 1, please mentioned the species name for Plasmodium? It should be also mentioned in the title.

In Table 1, please start with upper cases for alkaloid names

In Table 1, the “standards” must be replaced by “reference drug”

Author Response

Comments from the reviewers:

Comments and Suggestions for Authors

The paper on anti-Plasmodium activity of selected Amaryllidaceae alkaloids is very interesting and well-written. However, it needs a major revision as suggested below.

In 2.2. Amaryllidaceae Alkaloids, please indicate origin of country for the mentioned plant species as well as plant parts used for isolation. In the same section, “thorough” should be corrected as “through”.

We thank the reviewer for these remarks and have now incorporated this information in the manuscript.

2.4. NMR Spectra of Newly Developed Derivatives should be corrected as 2.4. NMR Spectra of Newly Synthesized Derivatives

Corrected.

Line 264, “Test concentrations were then incubated” should be modified as concentrations cannot be incubated. It might be suggested as “samples at tested concentrations”.

Corrected.

“3.1. Studied Alkaloids and Semisynthetic Derivatives” should be corrected as “3.1. Studied Alkaloids and Their Semisynthetic Derivatives”

Corrected.

Line 302, what is single tazettine type alkaloid? Indicate here please.

This was caused by a formatting error, for which we apologize, and has now been corrected.

In Figure 1, please mentioned the species name for Plasmodium? It should be also mentioned in the title.

The title and Figure 1 have been modified to include the Plasmodium species.

In Table 1, please start with upper cases for alkaloid names

Revised.

In Table 1, the “standards” must be replaced by “reference drug”

Revised.

Yours Sincerely,

Lucie Cahlíková, prof.

Lucie Cahlikova, Prof.; Dr., Department of Pharmaceutical Botany, Faculty of Pharmacy, Charles University, Heyrovskeho 1203, 500 05 Hradec Kralove, Czech Republic

Tel.: + 420 731 134 983

E-mail address: [email protected]

ORCID: 0000-0002-1555-8870

Hradec Králové, 19th, January, 2023

Reviewer 2 Report

Very interesting work; it is quite incredible that their precursor (Ambellin) has
an IC50 > 10 µM but that a simple benzoylation provides compounds
(28m and 28 n), with an IC50 around 50 nM... Congratulations to the authors !  

It is however unfortunate that the work « restricted » to Plamodium berghei infecting hepatoma cell line Huh-7. We are far from the real life (for example absence of CYP or non-representative cytochromes). This “easy” screen has to be completed by more appropriate screen: P. berghei in primary hepatocytes and P. falciparum in primary human hepatocytes. This screen can be “restricted” to both molecules 28m and 28n.

I have a few questions: 1.      IC50 of Primaquine. Primaquine does not seen to have been tested during their own experiments, which is of course regrettable. Where did they found this result? 2.      P. berghei has been cultivated in Huh-7. Why was the cytotoxicity performed in other cells, HepG2? These alcaloids (Polygonatum, Amaryllis, Convallaria majalis …) are in general very toxic. It is to be hoped that the work of chemists (remarkable in this work), will allow, if these molecules are confirmed as active, to neutralize this toxicity  

Author Response

Very interesting work; it is quite incredible that their precursor (Ambellin) has
an IC50 > 10 µM but that a simple benzoylation provides compounds
(28m and 28 n), with an IC50 around 50 nM... Congratulations to the authors !  

It is however unfortunate that the work « restricted » to Plamodium berghei infecting hepatoma cell line Huh-7. We are far from the real life (for example absence of CYP or non-representative cytochromes). This “easy” screen has to be completed by more appropriate screen: P. berghei in primary hepatocytes and P. falciparum in primary human hepatocytes. This screen can be “restricted” to both molecules 28m and 28n.

The results presented in this publication are the outcome of pilot research focused on AmA derivatives and their activity against the hepatic stage of P. berghei infection. The reviewer raises an interesting concern, that we are also well aware of. However, at this initial stage of compound screening for this stage of infection, it is customary to employ hepatoma cell lines as host hepatic cells, as amply documented (e.g. REFs…). The reviewer’s suggestion is well noted and will be the focus of future work, in which a series of more stable derivatives (ethers) corresponding to the most active substances will be synthesized and evaluated in more physiological models of hepatic infection, including primary hepatocytes and rodent models of liver infection.

I have a few questions:

  1. IC50 of Primaquine. Primaquine does not seen to have been tested during their own experiments, which is of course regrettable. Where did they found this result?

The activity of primaquine is routinely tested in the Prudêncio Lab and found to be highly reproducible. The IC50 value indicated has been obtained in the course of the remaining IC50 determinations reported in this manuscript and is in agreement with previously reported values REFs.

  1. P. bergheihas been cultivated in Huh-7. Why was the cytotoxicity performed in other cells, HepG2? These alkaloids (Polygonatum, Amaryllis, Convallaria majalis…) are in general very toxic. It is to be hoped that the work of chemists (remarkable in this work), will allow, if these molecules are confirmed as active, to neutralize this toxicity

Although cytotoxicity has been determined on the HepG2 cell line, this is also evaluated in the Huh-7 cell line during the assessment of the antiplasmodial activity of the compounds, at the concentrations employed in the initial screening and IC50 determination, by the AlamarBlue assay, since the potential hepatotoxicity of the tested compounds would be a serious adverse effect.

In light of these changes, we are positive that our revised manuscript meets the criteria to be published in Pharmaceuticsand would be of interest for all readers from the scientific community.

Yours Sincerely,

Lucie Cahlíková, prof.

Lucie Cahlikova, Prof.; Dr., Department of Pharmaceutical Botany, Faculty of Pharmacy, Charles University, Heyrovskeho 1203, 500 05 Hradec Kralove, Czech Republic

Tel.: + 420 731 134 983

E-mail address: [email protected]

ORCID: 0000-0002-1555-8870

Hradec Králové, 19th, January, 2023

Round 2

Reviewer 2 Report

Interesting work as a first screening, but I still believe that without confirmation of the results in an appropriate system (primary human hepatocytes and Plasmodium falciparum), the work is not achieved...

Author Response

Dear Editor,

we have received second comments to our manuscript: “Derivatives of Amaryllidaceae Alkaloid Ambelline as Selective Inhibitors of Hepatic Stage of Plasmodium berghei Infection In Vitro“ Manuscript ID: pharmaceutics-2155751. We tried to deal with the opponent's comments as best we could.

Reviewer #2

Interesting work as a first screening, but I still believe that without confirmation of the results in an appropriate system (primary human hepatocytes and Plasmodium falciparum), the work is not achieved...

We appreciate the reviewer's comment regarding the system employed to assess the hepatic stage antiplasmodial activity of these compounds. However, we respectfully argue that performing these experiments employing human primary hepatocytes and P. falciparum parasites is not only out of the scope of this publication but also outside our available budget and, in fact, impossible to perform within any reasonable amount of time. We respectfully draw the reviewers' attention to the following facts: human hepatoma cell lines and rodent Plasmodium parasites are the most commonly employed system to assess the hepatic stage activity of antiplasmodial compounds. This is extensively documented by work carried out in our laboratory (A.J.S. Alves, N.G. Alves, I. Bártolo et al. 2022, Front. Chem. DOI: 10.3389/fchem.2022.1017250; C.S. Pereira, H.C. Quadros, S.Y. Aboagye, et. al. 2022, Pharmaceutics, DOI: 10.3390/pharmaceutics14061251; E.A. Lopes, R. Mestre, D. Fontinha et al. 2022, Eur. J. Med. Chem, DOI: 10.1016/j.ejmech.2022.114324; D. Fontinha, F. Arez, I.R. Gal et al. 2022, ACS Inf. Dis., DOI: 10.1021/acsinfecdis.1c00640; N.G. Alves, I. Bártolo, A.J.S. Alves et al. 2021, Eur. J. Med. Chem, DOI: 10.1016/j.ejmech.2021.113439) and others (N. Sharma, Y. Gupta, M. Bansal et al. 2020, RSC Adv., DOI: 10.1039/D0RA03997G; J. Swann, V. Corey, A.A. Scherer et al. 2016, ACS Inf. Dis., DOI: 10.1021/acsinfecdis.5b00143; M.K.W. Mackwitz, E. Hesping, Y. Antonova-Koch et al. 2019, ChemMedChem, DOI: 10.1002/cmdc.201800808). Part of the reason why this is the system of choice for this kind of assays stems from the substantial difficulties in producing P. falciparum-infected mosquitoes, which are required for the generation of P. falciparum sporozoites (please see M. Prudêncio, M.M. Mota, A.M. Mendes 2011 “A toolbox to study liver stage malaria”, Trends Parasitol., DOI: 10.1016/j.pt.2011.09.004; for a full explanation about this). In fact, P. falciparum-infected mosquitoes are a resource that is only available in a handful of laboratories in Europe and around the world. The only alternative to the use of P. falciparum sporozoites obtained from mosquitoes is to purchase them in cryopreserved vials from the USA company Sanaria, Inc.. However, these are sold at a prohibitive price and, in any case, their delivery time would not be compatible with the time allowed for the current revision. Likewise, human primary hepatocytes are also not trivial to access. They can either be obtained through a protocol with a hospital that performs liver biopsies or liver transplants, which we don't have in place. Again, the alternative is to purchase cryopreserved human primary hepatocytes but, again, their price is not compatible with our budget and, in any case, there would be no guarantee of success because not all donors are easily infected by P. falciparum sporozoites in vitro. We trust the reviewer will understand that while performing these activity cells using human primary cells and human-infective parasites might sound ideal, this is not only completely impractical but also not common practice for a study such as the one in question.

In light of this explanation, we are positive that our revised manuscript meets the criteria to be published in Pharmaceutics and would be of interest for all readers from the scientific community.

Lucie Cahlíková, prof.
